# Land Use/Land Cover Change Analysis Using Multi-Temporal Remote Sensing Data: A Case Study of Tigris and Euphrates Rivers Basin

**Azher Ibrahim Al-Taei** [1,*], **Ali Asghar Alesheikh** [1] and **Ali Darvishi Boloorani** [2]

1   Faculty of Geodesy and Geomatics Engineering, K. N. Toosi University of Technology, Tehran 19967-15433, Iran
2   Department of Remote Sensing & GIS, Faculty of Geography, University of Tehran, Tehran 14155-4665, Iran
*   Correspondence: azher.altaei@email.kntu.ac.ir

**Abstract:** Multi-temporal land use/land cover (LULC) change analysis is essential for environmental planning and recourses management. Various global LULC datasets are available now. However, they do not show strong consistency on a regional scale and are mainly time limited. Therefore, high-quality multi-temporal LULC mapping with reasonable consistency on a regional scale is still demanding. In this study, using the Landsat 7, Landsat 8, and the NASA digital elevation model (DEM), LULC mapping of the Tigris and Euphrates rivers basin (TEB) was performed by random forest (RF) classifier in the Google Earth Engine platform during 2000–2022. The spectral bands, spectral indices, morphological, and textural features were applied in the developed procedure. The results indicated that the proposed approach had accurate performance (accuracy = 0.893 and an F score = 0.820) with a good consistency with previous studies. The feature importance evaluation was carried out using Gini index, and spectral indices were identified as the most important features in LULC mapping. Overall, severe LULC change has happened in the TEB during the last two decades. Our results revealed the expansion of water and built-up classes while trees class has experienced a decreasing trend. From a regional perspective, three main areas in the east and south-east of Iraq, north-west of Iraq, and east of Syria were identified where LULC change was intense. These areas are prone to land degradation and dust storms emission problems, and it is necessary to take steps to prevent severe LULC changes in them.

**Keywords:** land use/land cover change mapping; Tigris–Euphrates rivers basin; remote sensing

## 1. Introduction

Land is a vital natural resource for human life. Land use/land cover (LULC) is a cross-cutting environmental variable with significant applications in hydrological modeling, watershed management, natural disaster management, climate change studies, and land management [1]. Economic and industrial development, along with rapid and uncontrolled population growth, especially in developing countries in the late twentieth and early twenty-first centuries, have multiplied LULC change [2]. Human-induced LULC change led to adverse impacts on the land's anthropogenic and natural features at different spatiotemporal scales [3]. It made LULC change monitoring a critical requirement. Quantitative assessment of changes in LULC using time series satellite imagery is one of the most effective tools for understanding and managing land transformation [2].

LULC change analysis is usually required by planners and decision-makers for effective planning and management interventions on local, national, regional, and even global scales [4]. Remote sensing (RS) data has been applied in various fields, including LULC monitoring [5]. As satellites provide data with different spatial and temporal resolutions, they are suitable tools for collecting LULC information and detecting temporal changes [6].

Detection and modeling of the LULC change based on multi-temporal RS data is a complex process. Conventional approaches to creating LULC maps are limited when the study area is large, or with high variations in elevation and steep topography [1,4]. Machine learning (ML) algorithms have strong potential for combining new exploratory variables in LULC change modeling using RS data [7]. With the development of ML algorithms and increased access to satellite data, generating up-to-date and accurate LULC maps has become increasingly feasible [8]. Accordingly, it is crucial to understand how different ML algorithms perform for reliable estimation of LULC maps using RS data [1]. Table 1 presents the most recent ML applications in LULC mapping using RS data.

**Table 1.** ML applications in LULC mapping using RS data.

| Reference | Objectives | Data Sources | Algorithms | Results |
|---|---|---|---|---|
| [3] | Spatiotemporal change analysis of LULC and land surface temperature in the Suha watershed, north-western highlands of Ethiopia | Landsat 5 Landsat 7 Landsat 8 | SVM | Obtained high accurate LULC maps show the expansion of agricultural lands, barren lands, and built areas and the reduction of grazing and shrublands. |
| [8] | Performance evaluation of ML algorithms for LULC mapping | Landsat 8 Sentinel-2 | SVM ANN MLC MD | The outcomes indicated that SVM had the most accurate performance, and Sentinel 2 data was slightly more accurate than Landsat 8. |
| [9] | Monitoring LULC maps using different ML algorithms in the Munneru River basin, India | Landsat 8 Sentinel-2 | SVM RF CART | All classifiers performed with high accuracy, but RF outperformed both SVM and CART. |
| [10] | Detailed and automated classification of LULC using ML algorithms | MCD12Q1 Landsat 5 | RF CART | Classifiers were validated using data from US and Australia, and RF outperformed CART in both validations. |
| [11] | Quantifying spatiotemporal dynamics of LULC in Cox's Bazar district, Bangladesh | Landsat 4 Landsat 5 Landsat 8 | RF | RF had highly accurate predictions. Vegetation cover and urban settlements expanded. Water bodies and bare lands were decreased. |
| [1] | Evaluation of ML classifiers for LULC mapping in the snow-fed Alaknanda River basin, north-west Himalayan region, India | Landsat 8 | RT SVM MLC | Validation results indicated that both RT and SVM algorithms generated good overall accuracy and performed more precisely than the MLC algorithm. |
| [12] | Long-time series high-quality and high-consistency land cover mapping based in the Hihe River basin | Landsat ASTER | RF | RF achieved an average precision of about 90% for estimating a long-time series LULC dataset. |
| [13] | Assessing the performance of ML algorithms for classifying LULC in a boreal landscape in south-central Sweden | Sentinel-2 | SVM RF Xgboost DNN | The highest accuracy was produced by SVM, closely followed by Xgboost, RF, and DNN. |
| [14] | LULC mapping with advanced ML classifiers in Shiraz City, Iran | Landsat 8 | SVM CTree DFMLP | DFMLP outperformed the other two algorithms for pixel-based and object-based Landsat 8 imagery classification. |
| [2] | Evaluation of ML algorithms for LULC classification for satellite observations | Landsat 4 Landsat 5 Landsat 8 | SVM RF ANN fuzzy ARTMAP SAM MHD | The accuracy assessment showed that all classifiers had similar accuracy levels with minor variation, but RF was the most accurate algorithm. |

Support vector machine (SVM), artificial neural network (ANN), maximum likelihood classification (MLC), minimum distance (MD), random forest (RF), classification and regression trees (CART), random trees (RT), extreme gradient boosting (Xgboost), deep neural network (DNN), complex tree (Ctree), derivative-free multi-layer perceptron (DFMLP), fuzzy adaptive resonance theory-supervised predictive mapping (fuzzy ARTMAP), spectral angle mapper (SAM), mahalanobis distance (MHD).

Although ML-based techniques have been attempted to develop LULC maps with high levels of accuracy [4] (Table 1), they still have challenges in LULC modeling [7] and must be evaluated in different land and climatic conditions [2]. However, among different implemented ML algorithms, RF has performed chiefly accurately. In addition, Landsat satellite imagery can be considered a suitable data source for multi-temporal LULC change analysis, as it provides repetitive earth observations with high spatial resolution. Accordingly, Landsat satellite imagery and the RF algorithm were chosen for this study.

Many global LULC maps are available based on different methods [15]. While widely used, due to the limitations of their spatial and temporal resolutions, many significant LULC changes are difficult or impossible to detect with them [16]. These global LULC maps usually show strong consistency on a global scale but have major deviations at

regional scale. Accordingly, high-quality and consistent multi-temporal LULC datasets are unavailable at regional scale [12].

The river basins play a crucial role in the environment and society, providing fresh water, regulating water flow and quality, protecting from natural hazards, and enabling biodiversity conservation. Accordingly, assessing the state of environmental management and natural resources using LULC maps is essential for any river basin or watershed [1]. The Tigris and Euphrates Rivers basin (TEB) is a large river basin in west Asia suffering from climate change and human activities [17,18]. Although the TEB situation affects people's lives in several countries, to our knowledge, few papers have addressed this region's multi-temporal LULC change analysis. In this regard, the primary purpose of this study was to create a multi-temporal LULC change analysis in the TEB. First, the RF classifier was adopted and evaluated for LULC mapping from 2000 to 2022. Then, created maps were analyzed to determine LULC changes.

## 2. Materials and Methods

### 2.1. Study Area

The TEB (Figure 1), in the Middle East, is one of the largest river basins with a long-term annual rainfall of less than 166 mm. The TEB expands between $36°41'$ to $52°8'$ E and $27°13'$ to $40°23'$ N. The TEB is a transnational and international region with an area of 879,000 square kilometers. Climate projections have predicted declining rainfall, declining river flow, and rising temperatures in this region [19].

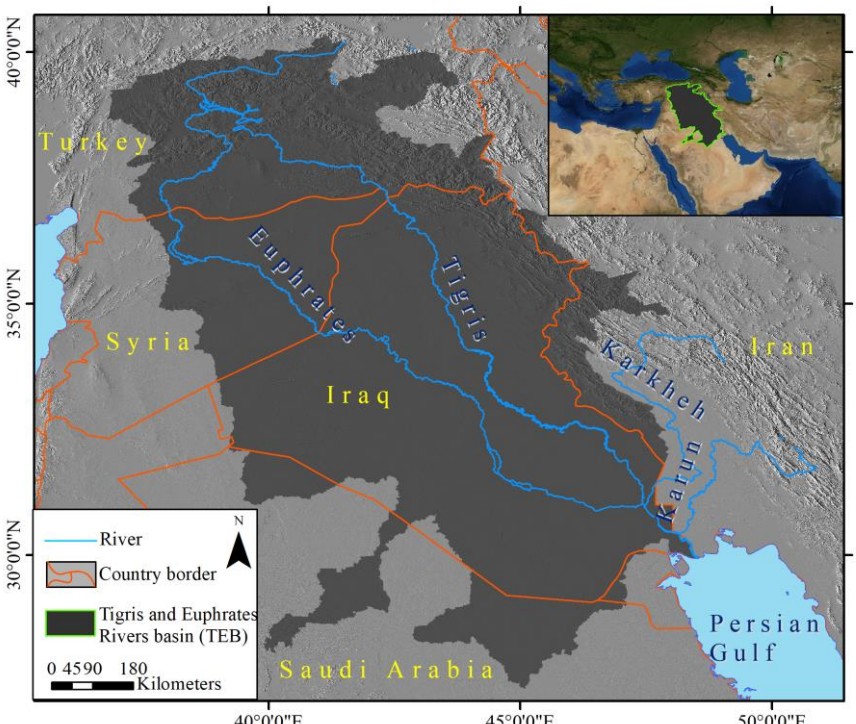

**Figure 1.** Location map of the TEB.

### 2.2. Dataset

Supervised classification algorithms require sufficient and representative training samples. These samples have a class label (LULC class) and modeling features (effective factors). After reviewing the literature [1,3,9,10,12,20], modeling features were selected to train the algorithm. Four morphological features, including elevation, aspect, hill-shade, and slope, were derived from NASA-DEM 30 m. The TEB is a vast area with varied morphology, and morphological features are crucial for such an area [20]. Nine modeling

features, including four spectral indices and six spectral bands (B, G, R, NIR, SWIR 1, and SWIR 2), were obtained from Landsat satellite images.

Spectral indices increase the information and enhance classification accuracy to map LULC [21,22]. Accordingly, the normalized difference built-up index (NDBI), the bare soil index (BSI), the normalized difference vegetation index (NDVI), and the modified normalized difference water index (MNDWI) were computed as modeling features. BSI is used to discriminate between bare soil and other LULC classes [23], and classifies bare soil areas [24]. The spectral characteristics of built-up areas are similar to bare soil, mapping built-up areas is a challenging task, and NDBI is an effective remotely sensed index for built-up mapping. NDVI is a standard vegetation index, as it can measure vegetation health and coverage [20,25]. MNDWI is calculated to enhance water features, and reduces built-up area features that are usually correlated with water in other indices [26]. The slope, aspect, NDBI, BSI, NDVI, and MNDWI features were obtained from the Google Earth Engine.

### 2.3. Methodology

As shown in Figure 2, the research steps are as follows:

1.  Preparation of data layers, including Landsat 8 (2013–2022) and Landsat 7 (2000–2012) images and the NASA-DEM.
2.  After filtering the cloud band, dates, and total cloud cover, six spectral bands, four spectral indices, and 18 textural features were extracted from the annual Landsat images. Each year, the median of Landsat images with a total cloud cover of less than 50% taken from April to September and considered the annual image. Landsat 7 gap filling was performed before feature extraction. To extract the textural features, the images were converted into gray-level 16-bit images with a linear weighted combination ($0.3 \times R + 0.59 \times G + 0.11 \times B$) [20]. The gray-level co-occurrence matrix (GLCM) algorithm was applied to extract their textural features. Elevation, aspect, hill-shade, and slope features were extracted from the NASA-DEM.
3.  Feature selection was performed using the Gini index.
4.  A total of 50,000 random sample points were selected and labeled in the study area by visual inspection using Google Earth Pro.
5.  The RF classifier was trained using 80% of the dataset. The remaining 20% of the dataset was used to assess the accuracy of the results using the confusion matrix.
6.  After accuracy assessment, the multi-temporal LULC maps were created. The LULC map for each year was estimated using training the model with samples from that year.
7.  The Modified Mann–Kendall (MMK) test was applied to determine the LULC classes change uniform trend, and a fifth polynomial was fitted to the LULC classes' data series to plot the LULC change trend.
8.  LULC maps were temporary analyzed to calculate the LULC changes.

### 2.4. Land Use/Land Cover (LULC) Classification

Generally, two kinds of methods have been applied for LULC classification: object-based and pixel-based methods. Object-based methods apply unsupervised classification algorithms to cluster unlabeled pixels of an image. Then, experts label some of these clusters to train supervised methods for classifying all clusters. Pixel-based methods use supervised classification algorithms to determine the LULC class of image pixels based on the labeled samples.

RF is a supervised ensemble ML algorithm that uses a group of decision trees, in which each tree creates its data model based on random sample data and random features. Then, the final prediction is chosen by averaging or voting between the trees' predictions [27]. To enhance the prediction ability of RF, the association between the trees should be reduced, and their intensity contributions should be increased [28].

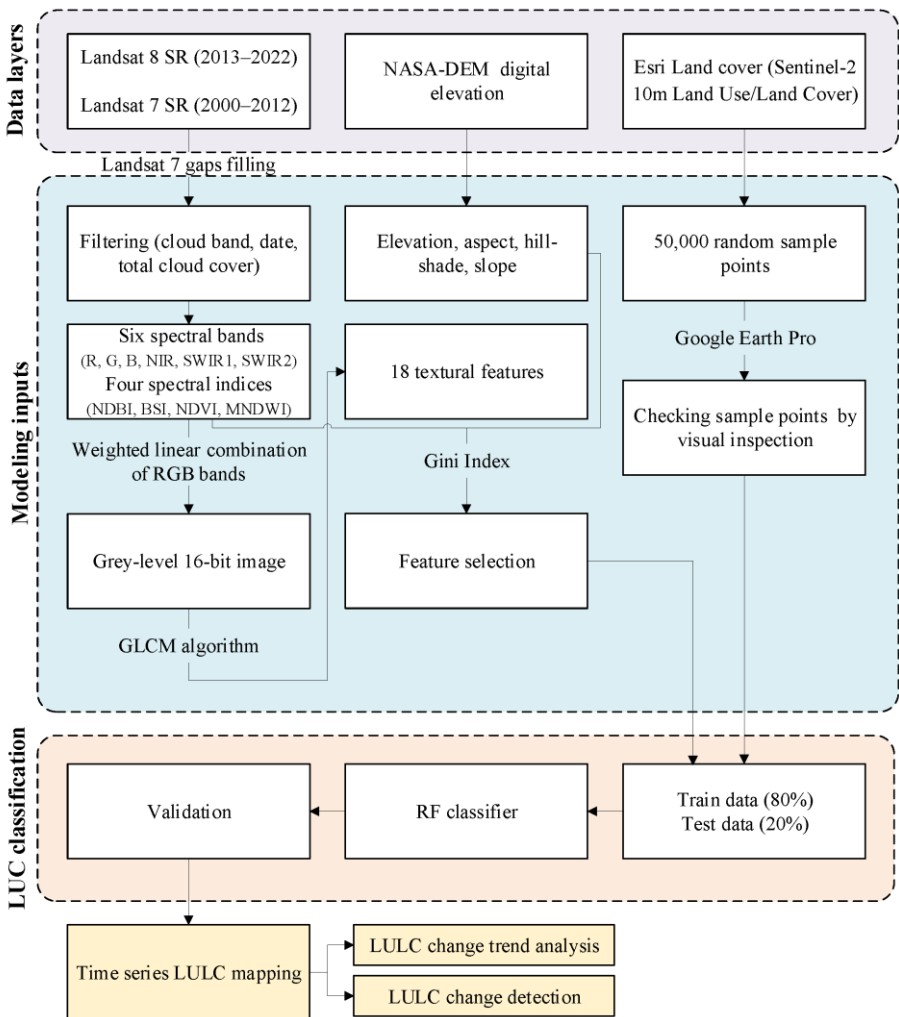

**Figure 2.** Developed methodology for multi-temporal land use/land cover change analysis in the Tigris and Euphrates rivers basin.

### 2.5. Classification Accuracy Assessment

The confusion matrix is a well-known performance measurement for ML-based RS classification algorithms. As Figure 3 shows, the true positive (TP) and true negative (TN) are the numbers of positive and negative samples correctly predicted by the classifier, respectively. False positive (FP) and false negative (FN) are the numbers of negative and positive samples incorrectly predicted by the classifier, respectively. Accordingly, the accuracy and f-score of the results were calculated by Equations (1) and (2), respectively [29].

$$\text{Accuracy} = \frac{\text{TP} + \text{TN}}{\text{TP} + \text{TN} + \text{FP} + \text{FN}} \tag{1}$$

$$\text{F} - \text{score} = \frac{2 \times \text{Precision} \times \text{Recall}}{\text{Precision} + \text{Recall}} \tag{2}$$

|  | Actually positive | Actually negative |
|---|---|---|
| Predicted positive | TP | FP |
| Predicted negative | FN | TN |

**Figure 3.** Confusion matrix [30].

## 3. Results

### 3.1. Classification Results

The TEB is a vast area containing various LULC classes. Tree-based ML algorithms are more practical for modeling big datasets for large-scale classification, as their architecture prevents tree growth from covering noisy outlier samples [31]. We used the Google Earth Engine platform to create RF models. The model contained 100 trees by default. The modeling inputs were features derived from Landsat 7, 8, and NASA-DEM, and the modeling targets were the LULC classes. Before training models, feature selection was performed using the Gini index to reduce the number of modeling features and improve the training process. The results of the feature selection are listed in Table 2. Accordingly, 18 features were selected for LULC modeling.

**Table 2.** Feature selection results performed by the Gini index. M (morphological feature), SB (spectral band), SI (spectral index), T (textural feature).

| Feature | Type | Mean Importance Weight | Selection | Feature | Type | Mean Importance Weight | Selection |
|---|---|---|---|---|---|---|---|
| Slope | M | 43.111 | True | Texture_shade | T | 25.838 | True |
| B | SB | 36.914 | True | SWIR 1 | SB | 25.776 | True |
| NDBI | SI | 35.369 | True | Aspect | M | 24.216 | False |
| BSI | SI | 34.832 | True | Texture_contrast | T | 24.025 | False |
| MNDWI | SI | 34.322 | True | Texture_dvar | T | 22.383 | False |
| NDVI | SI | 32.167 | True | Texture_corr | T | 21.882 | False |
| Texture_savg | T | 32.111 | True | Texture_idm | T | 21.824 | False |
| NIR | SB | 31.928 | True | Hillshade | M | 21.798 | False |
| Elevation | M | 31.610 | True | Texture_var | T | 21.129 | False |
| SWIR 2 | SB | 29.811 | True | Texture_imcorr1 | T | 17.162 | False |
| Texture_svar | T | 28.954 | True | Texture_dent | T | 10.376 | False |
| Texture_inertia | T | 28.890 | True | Texture_sent | T | 10.343 | False |
| Texture_diss | T | 27.843 | True | Texture_imcorr2 | T | 8.945 | False |
| R | SB | 27.032 | True | Texture_ent | T | 7.484 | False |
| G | SB | 26.987 | True | Texture_asm | T | 6.696 | False |
| Texture_prom | T | 26.973 | True | Texture_maxcorr | T | 0.000 | False |

To make training samples, random sample points were identified in the study area, and seven LULC classes of water, trees, flooded vegetation, crops, built area, bare ground, and rangeland were considered. We followed Esri Land Cover dataset to define each LULC class as follows:

- Water: areas where water was mainly present throughout the year.
- Trees: notable areas of tall, dense vegetation.
- Flooded vegetation: areas where vegetation and water are mixed mainly throughout the year.
- Crops: crops, cereals, and grasses planted by humans at lower tree height.

- Built area: impervious surfaces and structures made by humans.
- Bare ground: areas of rock or soil containing very poor vegetation.
- Rangeland: areas that are covered by homogenous grasses.

Sample points were overlaid with Google Earth Pro images. They were then labeled according to the classes' definitions. While sample points were homogeneously distributed, their number for each class was defined by the area ratio of the class to the whole area of the TEB. For each year, the classifier trained with randomly selected 80% of samples (train data) and validated with the remaining 20% of samples (test data). We averaged the validation results from 2000 to 2022. The mean accuracy of train data, mean accuracy of test data, mean F-score of train data, and mean F-score of test data with slight standard deviation (SD) values were 0.995, 0.893, 0.993, and 0.820, respectively. After producing the LULC maps, they were processed by the majority filter to increase their generalizability. Figure 4 shows the created LULC maps.

### 3.2. Land Use Land Cover (LULC) Change Analysis

To understand whether there is a uniform trend in LULC change of the TEB, the Modified Mann–Kendall (MMK) test was applied to multi-temporal LULC maps. We used the MMK test proposed by [32] to deal with data autocorrelation. It uses a variance correction approach to improve trend analysis. In this statistical test, H0 and H1 were considered the absence and existence of uniform trends in the data series, respectively. The results are summarized in Table 3. A uniform increasing trend for the built area class and a uniform decreasing trend for the trees class were detected. In addition to the MMK test, we fitted the fifth polynomial to each LULC class data series (Figure 5). Although the MMK test indicated no uniform trend for flooded vegetation, crops, bare ground, and rangeland exists, the trend lines shown in Figure 5 illustrate that water, flooded vegetation, crops, and rangeland classes will decrease. In contrast, trees, built areas, and bare ground will expand in the coming years.

**Table 3.** The MMK test results.

| LULC Class | H0 | H1 | Trend Type | *p* Value |
|---|---|---|---|---|
| Water | Rejected | Accepted | No trend | 0.291 |
| Trees | Rejected | Accepted | Decreasing | 0.006 |
| Flooded vegetation | Accepted | Rejected | No trend | 0.291 |
| Crops | Accepted | Rejected | No trend | 0.117 |
| Built Area | Rejected | Accepted | Increasing | 0.000 |
| Bare ground | Accepted | Rejected | No trend | 0.908 |
| Rangeland | Accepted | Rejected | No trend | 0.422 |

We compared the estimated LULC maps to carry out the LULC to mark points (areas) where LULC changed during 2000–2022. These points were then processed by the kernel density estimation (KDE) method, and the output was classified into two classes. Accordingly, pixels with a value of more than $2.5 \times STD$ were considered outliers where LULC change was highly severe. It was found that the east, south-east, and north-west of Iraq and east of Syria have faced more LULC change in recent years (Figure 6).

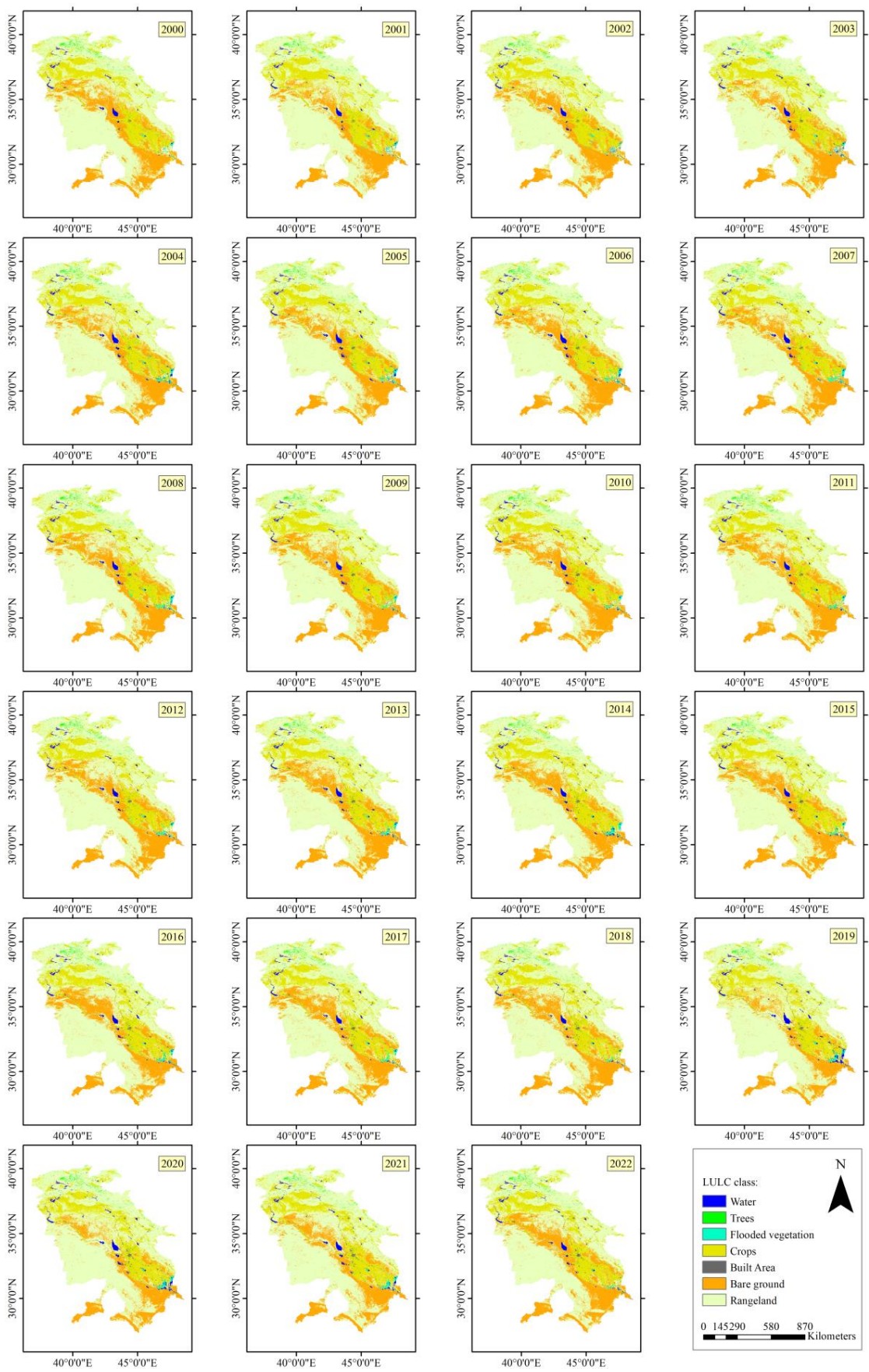

**Figure 4.** Multi-temporal LULC maps of the TEB during 2000–2022.

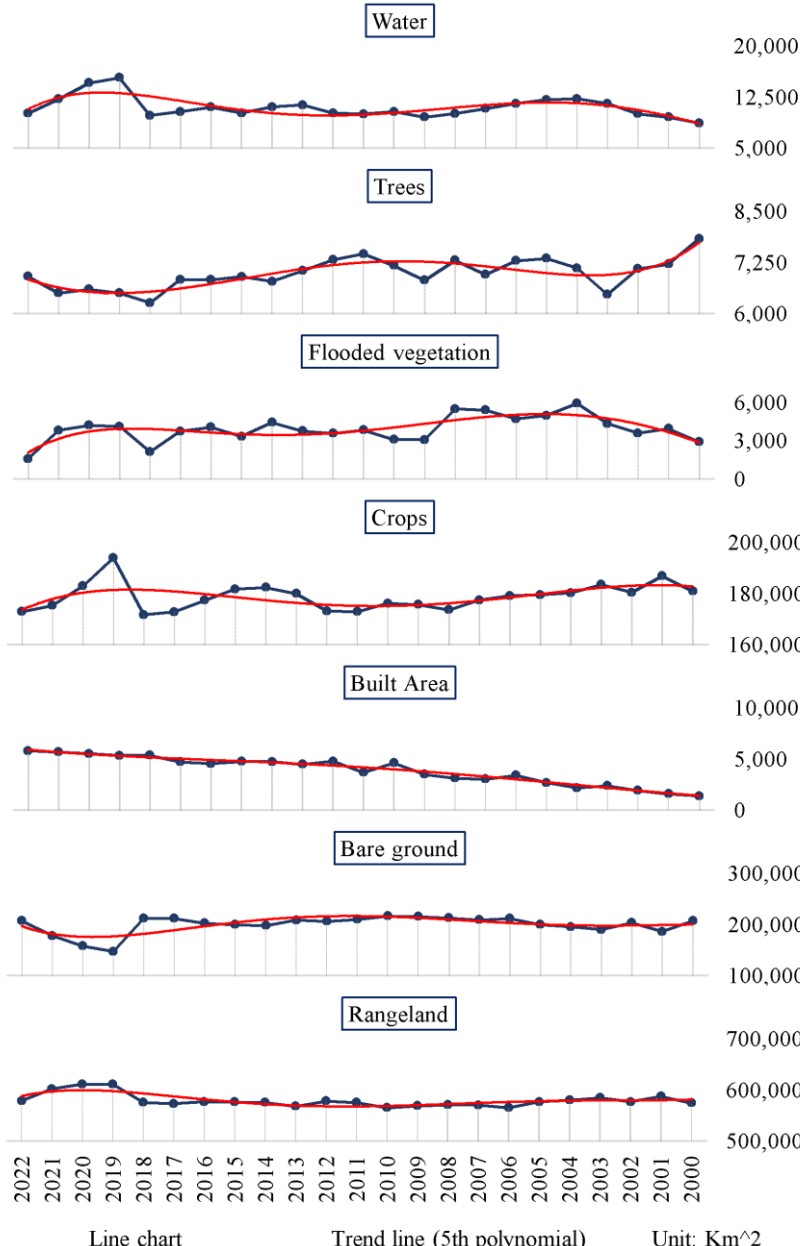

**Figure 5.** Fitting a fifth polynomial function to LULC classes' data series.

Pearson correlation coefficients were calculated to determine the linear relationship between LULC classes (Figure 7). It is a statistical criterion that has been widely applied as a measurement of relationships between two variables. It has a value between −1 and +1, where −1 indicates the negative relationship and the value of +1 means the positive relationship [33]. Investigating the relationship between LULC classes using Pearson correlation coefficients allows us to determine how the changes in an specific LULC class would impact the change in another LULC class. Our results revealed that water class had high anti-correlation with the bare ground (−0.823), moderate anti-correlation with trees (−0.458), high correlation with rangeland (0.734), and moderate correlation with crops (0.560). Trees class was associated with the bare ground (0.446) and dissociated with the built area (−0.578) and rangeland (−0.488). Flooded vegetation had moderate anti-correlation (−0.401) with the built area. Crop class had high anti-correlation with the bare ground (−0.732) and was associated with rangeland (0.539). Bare ground was highly anti-correlated with rangeland (−0.960). No more considerable correlation was observed.

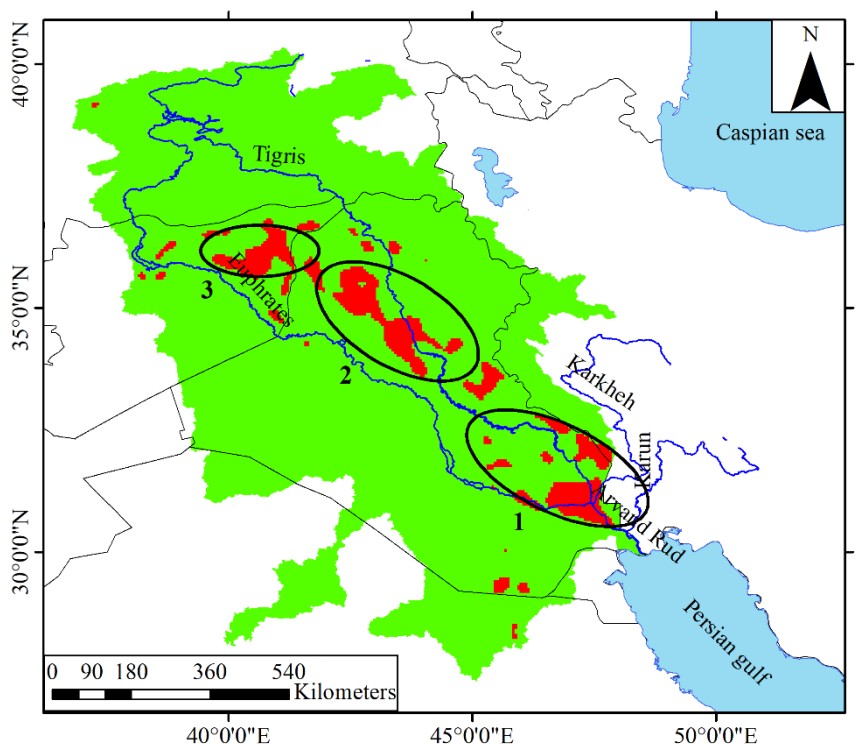

**Figure 6.** Three main geographical zones of the TEB where LULC has changed significantly during 2000–2022. (1) East and south-east of Iraq, (2) north-west of Iraq, (3) east of Syria.

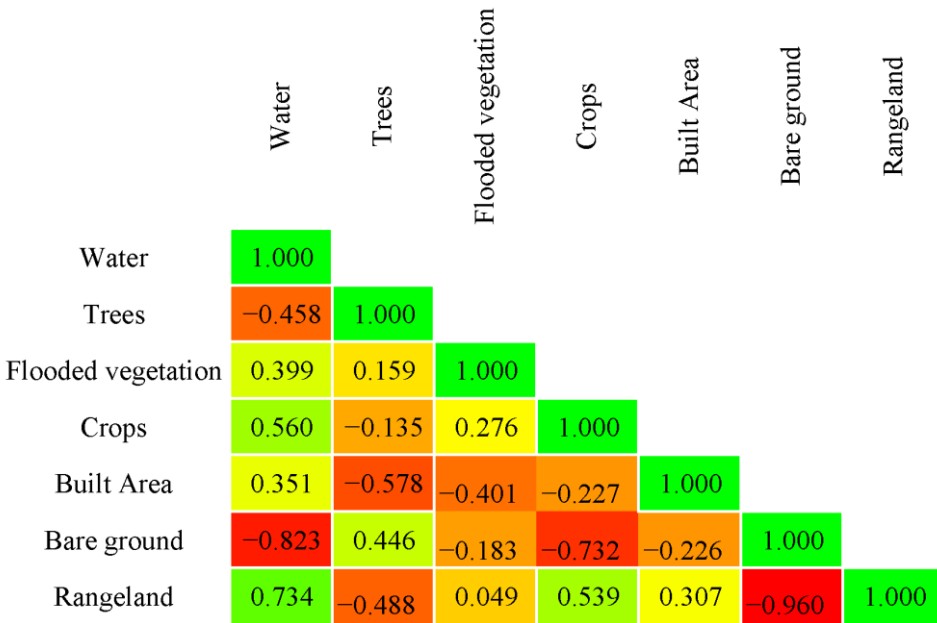

**Figure 7.** Pearson correlation coefficients between LULC classes.

## 4. Discussion and Conclusions

Many papers (Table 1) have investigated different ML-based approaches for LULC classification, and reached an overall accuracies between 0.73–0.95. While the TEB is a great river basin in west Asia, and LULC change analysis is essential to support future planning, management, and sustainable development of it and the surrounding areas, there are limited studies conducted on the TEB to analyze the LULC changes. LULC change analysis is applicable for numerous studies for every river basin, including ecosystem

conservation and services, sustainable land use planning, natural disaster prevention, and water security [34].

In this study, due to computational power limitation, LULC in the TEB was classified using RF without utilizing any hyper-parameter tuning methods, and a good overall accuracy of 0.893 was achieved. Some studies applied hyper-parameter tuning methods to boost the accuracy of LULC classification [8,11,13]. Hyper-parameter tuning is an essential process for ML-based modeling [35]. However, compared with other ML classifiers, RF is less influenced by hyper-parameters [36].

Based on slight SD values [37], the RF classifier had excellent generalizability. Accordingly, LULC maps in different years (2000–2022) and using different data (Landsat 7 and Landsat 8 images) with an acceptable level of accuracy were created. Although created LULC maps may contain uncertainties due to misclassification, homogeneity, and autocorrelations within train samples, the approach used in this research was easy, cheap, and fast, and provides beneficial information in less developed areas, such as the TEB, where there is a lack of data for resource management.

Using limited features in the LULC classification process may cause weak results in areas with heterogeneous landscape characteristics like the TEB. To overcome this issue, morphological, spectral indices, and textural features are usually utilized along with spectral bands to enhance classification accuracy [38]. On the other hand, many features can disturb the training process of ML algorithms. Therefore, we prepared various features for LULC classification and performed feature selection using the Gini index, and 18 features out of 32 available features were applied.

The Gini index indicated that, on average, spectral indices were the most important feature type. This was expected as spectral indices are defined to increase information and improve modeling. Slope and elevation were two of the most important features. The TEB is an area with various morphological conditions. Accordingly, these two features were highly diverse and enhanced LULC classification results. However, the aspect and hill-shade were not highly varied and had lower importance. We found that the blue band was the most important spectral band in modeling, followed by NIR, SWIR 2, red, green, and SWIR 1. However, different researchers evaluated the usefulness of spectral bands for LULC classification in different study areas and reached different results [39–41]. Accordingly, LULC classes and conditions of the study area should be considered when choosing the most suitable spectral bands for LULC classification.

MMK test results showed that only trees and built area classes had uniform trends. These findings are consistent with previous works that studied LULC change in Iraq and Syria. Since urbanization in Iraq generally has a vague strategy, and there is no proper urban policy [42], rapid urban expansion is one of the leading environmental problems in this country [43–45]. In addition, the loss of orchards due to reasons like uncontrolled LULC change [46–50], rising temperatures, and increased water salinity [51] is an issue that Iraq is struggling with. There is a similar situation in Syria, where urban areas in eastern governorates were expanded due to population growth [52], and analyzing NDVI showed a decrease in green areas [53] in recent years.

According to some observations of southern Iraq (Basra city) [54], eastern Iraq [55], the Diyala River basin [56], and national-scale data of Iraq [57], precipitation in the TEB sharply increased in 2019 compared to previous years, and then decreased quickly. These observations are consistent with research findings where a significant increase in water class was observed in 2019 (the exceptional wet year with abundant spring rains [55]), and then it started to decrease. Other LULC classes correlated or anti-correlated with water, followed the water class trend.

The trees class was anti-correlated with the water and rangeland classes. It is expected that increasing water class leads to an increase in trees class. However, while rapid built area expansion was indicated in the TEB, some observations state that urban area expansion has increased the reduction and fragmentation of orchards and trees in some parts of the TEB, such as the Iraqi Kurdistan [46], northern Iraq [47], Kirkuk [48], Karbala [49], and

Baghdad [50] governorates. Similarly, the trees class was anti-correlated with the built area. Crops had a moderate correlation with water, high anti-correlation with bare ground, and moderate correlation with rangeland. Vegetation and green cover classes contrast with bare ground. In other words, the findings indicate that the more water, the more crops, vegetation, and greenery, and the less bare ground.

Rapid and drastic LULC change severely affects the environment and human life. We found that the most severe LULC change in the TEB occurred in the east, south-east, and north-west of Iraq and in east of Syria, while previous findings defined these areas as dust prone sources [58]. Changes in precipitation and water flow may increase dissolved substances of water bodies and harm aquatic habitats and freshwater supplies [59]. Reduction of vegetation and greenery in arid and semi-arid regions such as the TEB are highly correlated with dust events [60]. Losing trees can lead to climate change, soil erosion, flooding, etc. Similarly, several papers have illustrated urban expansion's effects on climate change [61–63]. In conclusion, it is essential to take specific steps to deal with the rapid LULC change in the TEB. For this purpose, future works can help prevent these consequences by investigating its influential factors. In addition, analyzing the current consequences of rapid LULC change in the TEB helps motivate decision-makers to take the necessary actions.

**Author Contributions:** Conceptualization, A.D.B.; methodology, A.D.B.; software, A.I.A.-T; validation, A.A.A.; formal analysis, A.I.A.-T.; investigation, A.D.B.; data curation, A.I.A.-T. and A.D.B.; writing—original draft preparation, A.I.A.-T.; writing—review and editing, A.A.A. and A.D.B.; visualization, A.I.A.-T.; supervision, A.A.A.; project administration, A.A.A. All authors have read and agreed to the published version of the manuscript.

**Funding:** This research received no external funding.

**Data Availability Statement:** The datasets used and/or analyzed during the current study are available from the corresponding author on reasonable request.

**Acknowledgments:** We thank the editor and the anonymous reviewers for their excellent suggestions to improve the original draft of the manuscript.

**Conflicts of Interest:** The authors declare no conflict of interest.

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
