# Peer review of "Land Use/Land Cover Change Analysis Using Multi-Temporal Remote Sensing Data: A Case Study of Tigris and Euphrates Rivers Basin"

_land, doi:10.3390/land12051101_

Round 1
Reviewer 1 Report
In this paper, the author attempts to use multi-source multi-temporal remote sensing data to map long-term land use/land cover for the Tigris and Euphrates river basins, and analyze land cover changes from 2000 to 2022. The results show that effective features are able to extract long-term land use/land cover information. The study provides some ideas worth referencing. Besides, there are some issues that need to be solved according to the following suggestions.
1. There are now many global LULC data, based on different methods. Therefore, the strengths of the study should be emphasized in comparison to these products. In addition, comparison experiments should be supplemented, including local detail map comparisons and accuracy indicators comparisons.
2. Unsupervised classification is used for pre-processing. Specifically, this content needs to be explained. Meanwhile, how to ensure the accuracy of the sample?
3. Could you explain whether the prediction results for other years are obtained by training the model with samples from only one year? Differences in long-term images need to be considered. Whether the images of different years need to be processed or corrected.
4. In line 236, “it was found that the RF classifier was not dependent on the distribution of train and test samples.” This sentence can be misleading, it is recommended to delete it.
5. There are multiple repeated descriptions of experimental parameters, such as 50,000 samples and the sample proportion. It is recommended to reduce redundant content and accurately and clearly express the information.
6. In line 157, “The modeling inputs were derived from Landsat 7 and Landsat 8 images, and NASADEM data, and modeling targets were LULC classes”, the input to the model should be features instead of images.
Author Response
Manuscript Number: land-2336876
Response to Reviewers
Dear Reviewer,
We appreciate the time and effort you dedicated to providing feedback on our manuscript and are grateful for the insightful comments and valuable improvements to our paper. We have incorporated most of the suggestions made by you. Those changes are highlighted within the manuscript. Please see below, in blue, for a point-by-point response to the reviewers' comments and concerns.
Comments from Reviewer #1:
- There are now many global LULC data, based on different methods. Therefore, the strengths of the study should be emphasized in comparison to these products. In addition, comparison experiments should be supplemented, including local detail map comparisons and accuracy indicators comparisons.
Response: Agree.
- The following part is added to the Introduction section to emphasize the strengths of the study in comparison to global LULC data:
Lines 69-73: Many global LULC maps are available based on different methods [15]. While widely used, due to the limitation of their spatial and temporal resolutions, many significant LULC changes are difficult or impossible to detect using them [16]. These global LULC maps usually show strong consistency at the global scale but have major deviations at the regional scale.
- Unfortunately, we had no access to detailed local maps, and this comparison experiment was impossible for us.
- A comparison between accuracy indicators is added to the discussion section as follows:
Lines 241-247: Previous works (Table 1) used different ML-based approaches and reached overall LULC classification accuracy of about 0.73 - 0.95. Some studies also applied hyper-parameters tuning methods to boost the accuracy of LULC classification [8,11,13]. Hyper-parameters tuning is an essential process for ML-based modeling [32]. However, we did not perform any hyper-parameters tuning method due to computational power limitation and could achieve a good overall accuracy of 0.893. This was because, com-pared with other ML classifiers, RF is less influenced by hyper-parameters [33].
- Unsupervised classification is used for pre-processing. Specifically, this content needs to be explained. Meanwhile, how to ensure the accuracy of the sample?
Response: We explained the mentioned part as follows:
Lines 139-144: Generally, two kinds of methods have been applied for LULC classification: object-based and pixel-based methods. Object-based methods apply unsupervised classification algorithms to cluster unlabeled pixels of an image. Then, experts label some of these clusters to train supervised methods for classifying all clusters. Pixel-based methods use supervised classification algorithms to determine the LULC class of image pixels based on the labeled samples.
- As mentioned in the methodology section, samples were labeled by visual inspection. We adopted this approach, which has been used successfully in LULC classification, for two reasons. First, we had no access to detailed local maps, and second, the study area was vast, and in situ sampling was impossible.
- Could you explain whether the prediction results for other years are obtained by training the model with samples from only one year? Differences in long-term images need to be considered. Whether the images of different years need to be processed or corrected.
Response: The prediction results for each year were obtained by training the model with samples from that year. This issue has not been explained in the text well. So, we added the following part:
Lines 128-130: The LULC map for each year was estimated by training the model with samples from that year.
- In line 236, “it was found that the RF classifier was not dependent on the distribution of train and test samples.” This sentence can be misleading, it is recommended to delete it.
Response: Agree. We rewrote the mentioned part as follows:
Line 248: Based on slight SD values [34], the RF classifier had excellent generalizability.
- There are multiple repeated descriptions of experimental parameters, such as 50,000 samples and the sample proportion. It is recommended to reduce redundant content and accurately and clearly express the information.
Response: Agree. We reviewed the manuscript again and tried to fix this problem.
- In line 157, “The modeling inputs were derived from Landsat 7 and Landsat 8 images, and NASADEM data, and modeling targets were LULC classes”, the input to the model should be features instead of images.
Response: Agree. The mentioned part is rewritten as follows:
Lines 166-167: The modeling inputs were features derived from Landsat 7, 8, and NASADEM data.

Reviewer 2 Report
I am delighted to review this manuscript entitled “Multi-temporal land use/land cover change analysis in Tigris and Euphrates rivers basin”. Specific comments are as follows:
1. Emphasis on big general ideas including originality and clearer contribution in relation to the literature: Making your research appealing to a broad audience is an important goal of this journal.
2. Related to my previous comment, please highlight whether the proposed econometric approach is best one matching with the economic question addressed in this paper. try to put out this issue in abstract as well as in the introduction.
3. Simple economic intuitions to explain the mechanisms that lead to the findings: Please make sure that the main finding is robust in explaining the mechanism of the impact with unambiguous supporting evidence.
4. Please provide as much practical insights/implications as possible that can attract a general reader to appreciate your finding and therefore can increase the citation of your work when published.
5. Forward-looking statements to motivate a comprehensive reading: The abstract needs to contain the key economic arguments and channels in a logical sequence. Also please try to make your abstract more accessible in a non-technical language.
6. Please revisit your title to make it self-explanatory and intuitively-appealing, reflecting the main idea of the paper in a concise and informative manner.
7. I read the paper carefully, I saw some language issues in the manuscript. Therefore, I recommend to proof read this paper from an English language expert.
Author Response
Manuscript Number: land-2336876
Response to Reviewers
Dear Reviewer,
We appreciate the time and effort you dedicated to providing feedback on our manuscript and are grateful for the insightful comments and valuable improvements to our paper. We have incorporated most of the suggestions made by you. Those changes are highlighted within the manuscript. Please see below, in blue, for a point-by-point response to the reviewers' comments and concerns.
Comments from Reviewer #2:
- Emphasis on big general ideas including originality and clearer contribution in relation to the literature: Making your research appealing to a broad audience is an important goal of this journal.
Response: Agree. We have incorporated most of the suggestions the reviewers made and believe that our manuscript's revised version can appeal to a broad audience.
- Related to my previous comment, please highlight whether the proposed econometric approach is best one matching with the economic question addressed in this paper. Try to put out this issue in abstract as well as in the introduction.
Response: Agree.
- The abstract is revised to address this comment.
- In the last two paragraphs of the introduction, we tried to answer why we selected this approach.
- Simple economic intuitions to explain the mechanisms that lead to the findings: Please make sure that the main finding is robust in explaining the mechanism of the impact with unambiguous supporting evidence.
Response: We tried to support our findings by comparing them with previous works. In this regard, we also used available data reports. However, as TEB is located in a less developed region, supporting evidence was limited.
- Please provide as much practical insights/implications as possible that can attract a general reader to appreciate your finding and therefore can increase the citation of your work when published.
Response: Agree. We tried to improve the discussion section and believe that the revised version of the manuscript can help readers in two different ways as follows:
- First, the accuracy of the proposed methodology was compared with the literature so that readers could use the methods in other works.
- Second, LULC change in the study area was analyzed, and findings were compared with previous works. In addition, the findings were reviewed from an environmental point of view. So, readers can consider our findings in future works.
- Forward-looking statements to motivate a comprehensive reading: The abstract needs to contain the key economic arguments and channels in a logical sequence. Also please try to make your abstract more accessible in a non-technical language.
Response: Agree. The abstract is revised according to this comment as follows:
Lines 10-25: Multi-temporal land use/land cover (LULC) change analysis is essential for environmental planning and recourses management. Various global LULC datasets are available now. However, they don’t show strong consistency at the regional scale and are mainly time limited. Therefore, high-quality multi-temporal LULC mapping with reasonable consistency in regional scales is still demanding. In this study, using the Landsat 7, Landsat 8, and NAS-ADEM digital elevation model, LULC mapping of the Tigris and Euphrates rivers basin (TEB) was performed by random forest (RF) classifier in the Google Earth Engine platform during 2000-2022. The spectral bands, spectral indices, morphological, and textural features were applied in the developed procedure. Results indicated that the proposed approach had accurate performance (accuracy = 0.893 and an F score = 0.820) and showed good consistency with previous findings. The Gini index evaluated feature importance, and spectral indices were the most important features in LULC mapping. Comparing the LULC maps revealed that water and built area classes expanded while trees classes decreased. Overall, severe LULC change has happened in TEB during the last two decades. In addition, three main areas in the east and southeast of Iraq, northwest of Iraq, and east of Syria were identified where LULC change was intense. These areas are prone to environmental problems, and it is necessary to take steps to prevent severe LULC changes in them.
- Please revisit your title to make it self-explanatory and intuitively-appealing, reflecting the main idea of the paper in a concise and informative manner.
Response: Agree. We changed the title according to this comment. The new title is as follows:
Lines 2-4: Land use/land cover change analysis using multi-temporal remote sensing data: A case study of Tigris and Euphrates Rivers basin
- I read the paper carefully, I saw some language issues in the manuscript. Therefore, I recommend to proof read this paper from an English language expert.
Response: Agree. We tried to fix this problem by reviewing the manuscript using Grammarly premium.

Reviewer 3 Report
Dear authors,
First of all, I congratulate you on your advanced work. However, in order to improve your presentation and contents you should work on the following aspects:
- Please indicate the sources of the information used in the figures.
- Include a location map at a smaller scale for contextualisation in Asia.
- Conclusions should be much more developed. It is recommended that it goes together with the discussion to take advantage of both contents.
The methods used are appropriate to the research. However, it would be important to indicate in detail all the sources used both in the text and in the figures produced as this will validate your method avoiding any kind of speculation. Furthermore, in order to contextualise the work, it would be interesting to include a location map of the study area relating it to a larger geographical space such as the Asian continent.
On the other hand, the conclusions are rather brief and could be expanded much further in relation to the results obtained. It is recommended that they go together with the discussion, so that each conclusion has its own debate and vice versa. In this way, it is possible to outline future lines of implementation of your work as well as future studies on this topic.
Overall, I think the study has a high impact on science, hence we reviewers do not reject it but demand these improvements so that it can be published. I encourage you to take these improvement actions and continue with this line of research.
Best regards
Author Response
Manuscript Number: land-2336876
Response to Reviewers
Dear Reviewer,
We appreciate the time and effort you dedicated to providing feedback on our manuscript and are grateful for the insightful comments and valuable improvements to our paper. We have incorporated most of the suggestions made by you. Those changes are highlighted within the manuscript. Please see below, in blue, for a point-by-point response to the reviewers' comments and concerns.
Comments from Reviewer #3:
- Please indicate the sources of the information used in the figures.
Response: We understand the importance of indicating the sources of the information used in the figures. However, all figures are created by authors using different software and are original. The only information we used was the boundary shape file of world countries and TEB, but no source was mentioned for them, and they are publicly accessible through the internet.
- Include a location map at a smaller scale for contextualization in Asia.
Response: Agree. According to this comment, Figure 1 is completely changed (Line 82).
- Conclusions should be much more developed. It is recommended that it goes together with the discussion to take advantage of both contents.
Response: Agree. The discussion and conclusion sections are combined in the revised manuscript. We also revised this section to enhance it.

Reviewer 4 Report
A solid, well written paper with an important message.
Figure 1: The (what I presume is) topographic scale is unlabeled.
Author Response
Manuscript Number: land-2336876
Response to Reviewers
Dear Reviewer,
We appreciate the time and effort you dedicated to providing feedback on our manuscript and are grateful for the insightful comments and valuable improvements to our paper. We have incorporated most of the suggestions made by you. Those changes are highlighted within the manuscript. Please see below, in blue, for a point-by-point response to the reviewers' comments and concerns.
Comments from Reviewer #4:
- Figure 1: The (what I presume is) topographic scale is unlabeled.
Response: Agree. The legend of Figure 1 was not appropriate. As can be seen in Line 82, Figure 1 is completely changed.

Round 2
Reviewer 1 Report
This paper has been revised under the review comments, and now clearly states the research purpose and significance. The authors have corrected inappropriate descriptions that may have existed before, enhancing the credibility and persuasiveness of the paper.
Author Response
Dear Reviewer,
We appreciate the time and effort you dedicated to providing feedback on our manuscript and are grateful for your encouraging comments on our paper.
Reviewer 2 Report
-
Author Response

(The authors gave the same response as above.)
